# Hospitalized patients dying with SARS-CoV-2 infection—An analysis of patient characteristics and management in ICU and general ward of the LEOSS registry

**Claudia Raichle** [1][☼]*, **Stefan Borgmann** [2][☼], **Claudia Bausewein** [3], **Siegbert Rieg** [4], **Carolin E. M. Jakob** [5], **Steffen T. Simon** [6], **Lukas Tometten** [5], **Jörg Janne Vehreschild** [5,7,8], **Charlotte Leisse** [5], **Johanna Erber** [9], **Melanie Stecher** [5,8], **Berenike Pauli** [6], **Maria Madeleine Rüthrich** [10], **Lisa Pilgram** [7], **Frank Hanses** [11,12], **Nora Isberner** [13], **Martin Hower** [14], **Christian Degenhardt** [15], **Bernd Hertenstein** [16], **Maria J. G. T. Vehreschild** [17], **Christoph Römmele** [18], **Norma Jung** [5]*, on behalf of the LEOSS Study Group[¶]

1 Department of Geriatric and Palliative Medicine, Tropenklinik Paul-Lechler-Krankenhaus, Tübingen, Germany, 2 Department of Infectious Diseases and Infection Control, Ingolstadt Hospital, Ingolstadt, Germany, 3 Department of Palliative Medicine, LMU Hospital, München, Germany, 4 Division of Infectious Diseases, Department of Medicine II, Medical Centre–University of Freiburg, Faculty of Medicine, University of Freiburg, Freiburg, Germany, 5 Department I of Internal Medicine, Faculty of Medicine and University Hospital of Cologne, University of Cologne, Cologne, Germany, 6 Department of Palliative Medicine, Faculty of Medicine, University of Cologne, Cologne, Germany, 7 Center for Internal Medicine, Medical Department 2, Hematology/Oncology and Infectious Diseases, University Hospital of Frankfurt, Frankfurt, Germany, 8 German Centre for Infection Research, partner site Bonn-Cologne, Cologne, Germany, 9 Department of Internal Medicine II, University Hospital rechts der Isar, Technical University of Munich, Munich, Germany, 10 Department of Internal Medicine II, Hematology and Medical Oncology, University Hospital Jena, Jena, Germany, 11 Emergency Department, University Hospital Regensburg, Regensburg, Germany, 12 Department for Infectious Diseases and Infection Control, University Hospital Regensburg, Regensburg, Germany, 13 Department of Internal Medicine II, Division of Infectious Diseases, University Hospital Würzburg, Würzburg, Germany, 14 Department of Internal Medicine, Klinikum Dortmund, Dortmund, Germany, 15 Department of Pharmacy, Städtisches Klinikum, Karlsruhe, Germany, 16 Klinikum Bremen-Mitte, Bremen, Germany, 17 Department of Internal Medicine 2, Infectious Diseases, University Hospital Frankfurt, Goethe University Frankfurt, Frankfurt am Main, Germany, 18 Department of Internal Medicine III–Gastroenterology and Infectious Diseases, University Hospital Augsburg, Augsburg, Germany

☼ These authors contributed equally to this work.
¶ Membership of the LEOSS Study Group is provided in the Acknowledgments.
* norma.jung@uk-koeln.de (NJ); raichle@tropenklinik.de (CR)

**Data Availability Statement:** Data cannot be shared publicly because of data protection reasons. Data are available from the LEAN European Open

## Abstract

### Background

COVID-19 is a severe disease with a high need for intensive care treatment and a high mortality rate in hospitalized patients. The objective of this study was to describe and compare the clinical characteristics and the management of patients dying with SARS-CoV-2 infection in the acute medical and intensive care setting.

### Methods

Descriptive analysis of dying patients enrolled in the Lean European Open Survey on SARS-CoV-2 Infected Patients (LEOSS), a non-interventional cohort study, between March

Survey on SARS-CoV-2 infected patients advisory board and Data Access/Ethics Committee (contact via analysis@leoss.net) for researchers who meet the criteria for access to confidential data.

**Funding:** The LEOSS registry was supported by the German Centre for Infection Research (DZIF) and the Willy Robert Pitzer Foundation Funding: Lisa Pilgram received a grant from DZIF (German Center for Infection Research) and Willy Robert Pitzer Foundation.

**Competing interests:** The authors have declared that no competing interests exist.

18 and November 18, 2020. Symptoms, comorbidities and management of patients, including palliative care involvement, were compared between general ward and intensive care unit (ICU) by univariate analysis.

## Results

580/4310 (13%) SARS-CoV-2 infected patients died. Among 580 patients 67% were treated on ICU and 33% on a general ward. The spectrum of comorbidities and symptoms was broad with more comorbidities ($\geq$ four comorbidities: 52% versus 25%) and a higher age distribution (>65 years: 98% versus 70%) in patients on the general ward. 69% of patients were in an at least complicated phase at diagnosis of the SARS-CoV-2 infection with a higher proportion of patients in a critical phase or dying the day of diagnosis treated on ICU (36% versus 11%). While most patients admitted to ICU came from home (71%), patients treated on the general ward came likewise from home and nursing home (44% respectively) and were more frequently on palliative care before admission (29% versus 7%). A palliative care team was involved in dying patients in 15%. Personal contacts were limited but more often documented in patients treated on ICU (68% versus 47%).

## Conclusion

Patients dying with SARS-CoV-2 infection suffer from high symptom burden and often deteriorate early with a demand for ICU treatment. Therefor a demand for palliative care expertise with early involvement seems to exist.

## Introduction

Since its first report in December 2019 in Wuhan, China, the Coronavirus disease 2019 (COVID-19) is spreading across the world [1–3]. Mortality rates for COVID-19 vary regionally but the WHO estimates that at least 3 million people have died globally from COVID-19 in 2020 [4–6].

Research focused mainly on the mode of action of the new virus and the course of disease for best therapeutic approaches to improve outcomes [7, 8]. However, detailed information about circumstances of dying with SARS-CoV-2 infection are limited. Keeley et al. describe in a rapid systematic review symptoms and clinical profiles in decedents but state a high demand for further information about end-of-life care [9].

Therefore, we aimed to describe the clinical characteristics and the end-of-life care of SARS-CoV-2 infected patients dying in intensive care units (ICU) and general wards with regard to their previous place of stay, symptom burden, comorbidities, involvement of specialist palliative care, and place of death.

## Methods

### Study design

Data were retrieved from the Lean European Open Survey on SARS-CoV-2 infected patients (LEOSS) registry, a multi-center non-interventional cohort study. Patients hospitalized and having died with laboratory-confirmed SARS-CoV-2 infection between March 18, 2020 and

November 18, 2020 were analysed in the present study. The date of the first positive PCR was defined as baseline. Reporting follows the STROBE guidelines [10].

## Participants and setting

We included patients with SARS-CoV-2 infection who have died and were treated in hospital on a general ward or an ICU.

## Data collection and processing

Study sites documented patients retrospectively in an electronic case report form using the online cohort platform ClinicalSurveys.net. To ensure anonymity in all steps of the analysis process, an individual LEOSS Scientific Use File (SUF) was created, which is based on the LEOSS Public Use File (PUF) principles described in Jakob et al. [11, 12]. Based on clinical and laboratory data, the SARS-CoV-2-infection was classified into an uncomplicated, complicated and critical phase at baseline and during the course of the disease. In case of dying on the day of the PCR being taken, they were classified as dead at baseline. Briefly, "uncomplicated" was mainly defined as oligo/asymptomatic, "complicated" by—amongst others—need of oxygen supplementation and "critical" by need for life supporting therapy. For more details, see Table 1 in the study of Rüthrich et al. [13].

We extracted data on the following characteristics: age, gender, place of stay before SARS-CoV-2 infection, month of diagnosis, phase at diagnosis, length of inpatient stay and ICU stay, symptoms and comorbidities, BMI, treatment on ICU, personal contacts with family and/or friends before or during the death phase, receiving palliative care before SARS-CoV-2 infection, involvement of a palliative care team during the inpatient stay, place of death (ICU, general ward), cause of death (dead from COVID-19, dead from other causes according to clinical judgement), premortal change of therapeutic goals from curative to palliative (progress of COVID-19, progress preexisting disease, other).

Binary variables were documented as yes, no, unknown or missing.

Symptoms and comorbidities occurring in at least 10% of the patients were analysed. The following symptoms were included: dry cough, productive cough, wheezing, dyspnea, nausea /emesis, muscle aches, muscle weakness, fever, delirium, excessive tiredness, headache and meningism. Presence of a symptom in at least one phase of the disease was counted as yes in our analysis.

The following 13 comorbidities were included in the analyses: arterial hypertension, atrial fibrillation, coronary artery disease, chronic heart failure, history of myocardial infarction, cerebrovascular disease, dementia, chronic kidney disease, acute kidney injury, diabetes mellitus with end organ damage, diabetes mellitus without end organ damage, COPD, solid tumor. Solid tumors with and without metastases were counted as one comorbidity.

Questions related to contacts, involvement of palliative care team and premortal change of therapeutic goals from curative to palliative could be answered with "quoted" or "not quoted". Blank answer boxes were counted as missings.

Values documented as unknown were defined as missing in the analysis. The number of missings differ as data collection has been adjusted throughout the project.

## Statistical analysis

We conducted descriptive analysis with absolute numbers and proportions of the extracted characteristics. Proportions in the ICU versus general ward group were analysed using the $chi^2$- test for five or more ordered categories and Fisher exact test for categorical data with four or less categories. As data were predominantly non-parametric, continuous data were analysed

**Table 1. Characteristics of patients dying with SARS-CoV-2 infection treated on ICU or general ward.**

| Characteristics | all patients | ICU | general ward | |
|---|---|---|---|---|
| | n (%) | n (%) | n (%) | p-Value |
| Gender | | | | |
| female | 181/580 (31) | 76/333 (23) | 71/166 (43) | **<0.0001** |
| male | 399/580 (69) | 257/333 (77) | 95/166 (57) | |
| Age | | | | |
| < 46 y | 10/580 (2) | 10/333 (3) | 0/166 (0) | **<0.0001** |
| 46–55 y | 28/580 (5) | 27/333 (8) | 1/166 (1) | |
| 56–65 y | 68/580 (12) | 64/333 (19) | 2/166 (1) | |
| 66–75 y | 124/580 (21) | 96/333 (29) | 17/166 (10) | |
| 76–85 y | 226/580 (39) | 111/333 (33) | 78/166 (47) | |
| > 85 y | 124/580 (21) | 25/333 (8) | 68/166 (41) | |
| Phase at diagnosis | | | | |
| uncomplicated | 179/572 (31) | 83/327 (25) | 74/165 (45) | **<0.0001** |
| complicated | 242/572 (42) | 126/327 (39) | 73/165 (44) | |
| critical | 128/572 (22) | 108/327 (33) | 11/165 (7) | |
| dead | 23/572 (4) | 10/327 (3) | 7/165 (4) | |
| Duration of inpatient stay (days) | n = 564 | n = 322 | n = 164 | |
| median (IQR) | 12 (6–22) | 17 (9–28) | 9 (6–16) | **<0.0001** |
| Number of comorbidities | | | | |
| 0 | 49/575 (9) | 41/333 (12) | 5/166 (3) | **<0.0001** |
| 1 | 88/575 (15) | 63/333 (19) | 14/166 (8) | |
| 2 | 106/575 (18) | 72/333 (22) | 23/166 (14) | |
| 3 | 118/575 (21) | 65/333 (20) | 34/166 (20) | |
| 4 | 88/575 (15) | 34/333 (10) | 39/166 (23) | |
| > = 5 | 116/575 (20) | 48/333 (14) | 47/166 (28) | |
| Most frequent comorbidities (> 10% prevalence) | | | | |
| arterial hypertension | 407/555 (73) | 225/320 (70) | 121/157 (77) | 0.128 |
| atrial fibrillation | 161/557 (29) | 74/319 (23) | 64/160 (40) | **0.0002** |
| coronary artery disease | 143/530 (27) | 72/310 (23) | 45/144 (31) | 0.0835 |
| chronic heart failure | 115/539 (21) | 52/314 (17) | 47/151 (31) | **0.0004** |
| history of myocardial infarction | 65/535 (12) | 28/314 (9) | 26/145 (18) | **0.0077** |
| cerebrovascular diseases | 93/548 (17) | 28/315 (9) | 38/156 (24) | **<0.0001** |
| dementia | 110/556 (20) | 19/316 (6) | 68/160 (43) | **<0.0001** |
| chronic kidney disease | 143/550 (26) | 65/315 (21) | 56/158 (35) | **<0.0008** |
| acute kidney injury | 75/506 (15) | 49/296 (17) | 23/139 (17) | 1.0 |
| diabetes without damage | 105/556 (19) | 66/319 (21) | 27/158 (17) | 0.3911 |
| diabetes with damage | 92/554 (17) | 44/314 (14) | 37/159 (23) | **0.014** |
| COPD | 76/555 (14) | 45/318 (14) | 21/159 (13) | 0.8883 |
| solid tumor with or without   metastases | 90/551 (16) | 45/315 (14) | 29/161 (18) | 0.2883 |
| Most frequent symptoms in any phase (> 10% prevalence) | | | | |
| Fever | 381/490 (78) | 235/288 (82) | 101/130 (78) | 0.23 |
| dyspnea | 366/478 (77) | 223/285 (78) | 86/121 (71) | 0.1282 |
| dry cough | 255/451 (57) | 160/267 (60) | 63/118 (53) | 0.2631 |
| excessive tiredness | 171/437 (39) | 80/252 (32) | 63/118 (53) | **<0.0001** |
| muscle weakness | 88/401 (22) | 40/235 (17) | 36/106 (34) | **0.0007** |
| delirium | 79/414 (19) | 40/245 (16) | 30/107 (28) | **0.0137** |
| productive cough | 73/407 (18) | 42/240 (18) | 23/104 (22) | 0.3682 |

(*Continued*)

**Table 1.** (Continued)

| Characteristics | all patients | ICU | general ward | |
| --- | --- | --- | --- | --- |
| | n (%) | n (%) | n (%) | p-Value |
| nausea / emesis | 61/408 (15) | 28/241 (12) | 24/103 (23) | **0.0082** |
| muscle ache | 50/395 (13) | 32/235 (14) | 16/100 (16) | 0.6101 |
| wheezing | 48/398 (12) | 27/236 (11) | 18/102 (18) | 0.1619 |
| headache | 44/387 (11) | 29/231 (13) | 8/96 (8) | 0.3396 |
| BMI | | | | |
| < 18.5 | 6/353 (2) | 2/233 (1) | 4/82 (5) | **<0.0001** |
| 18.5–24.9 | 109/353 (31) | 54/233 (23) | 34/82 (41) | |
| 25–29.9 | 133/353 (38) | 92/233 (39) | 34/82 (41) | |
| 30–34.9 | 64/353 (18) | 49/233 (21) | 8/82 (10) | |
| > 34.9 | 41/353 (12) | 36/233 (15) | 2/82 (2) | |

with the Mann-Whitney-U test. Reported p values are 2-sided and p<0.05 was considered statistically significant.

Age was documented and analysed in predefined categories of 10 years with patients < 46 years summarized in one category due to the low numbers of deaths in younger patients.

All data management and statistical analysis were conducted using GraphPad5.

## Ethical statement

Data were recorded completely anonymous and no patient-identifying data were stored. Written patient informed consent was waived. Approval for LEOSS was obtained by the applicable local ethics committees of all participating centers and registered at the German Clinical Trails Register (DRKS, No. S00021145).

## Results

Of 4310 SARS-CoV-2 infected patients, 580 (13%) were hospitalized and died. Data about ICU treatment were available from 499/580 (86%) patients. 333/499 patients (67%) received treatment on ICU compared to 166/499 (33%) on a general ward exclusively (Fig 1). 19/333 (6%) patients treated on ICU were transferred to a general ward before death.

97% of the cohort had been treated in Germany, the remaining 3% in Italy, Turkey, France, Spain, Switzerland, and Austria. 510/580 patients (88%) were infected in April 2020 or earlier.

### Characteristics of patients dying with SARS-CoV-2 infection

Patient characteristics and clinical data are outlined in Table 1.

Patients treated exclusively on the general ward had a significantly higher age distribution (>65 years: 98% (163/166) versus 70% (232/333)), suffered significantly from more comorbidities (≥ four comorbidities: 52% (86/166) versus 25% (82/333)), and had a shorter hospital stay (inpatient stay (median; IQR): 9 (6–16) versus 17 (9–28); p< 0.0001). 69% (393/572) of the patients were in an at least complicated phase at SARS-CoV-2 detection with a higher proportion of patients in a critical phase or dying the day of diagnosis treated on ICU (36% (118/333) versus 11% (18/165)). The spectrum of symptoms was broad with fever (381/490, 78%), dyspnea (366/478, 77%) and dry cough (255/451, 57%) most frequently observed. Patients on the general ward suffered significantly more often from excessive tiredness, muscle weakness and nausea/emesis.

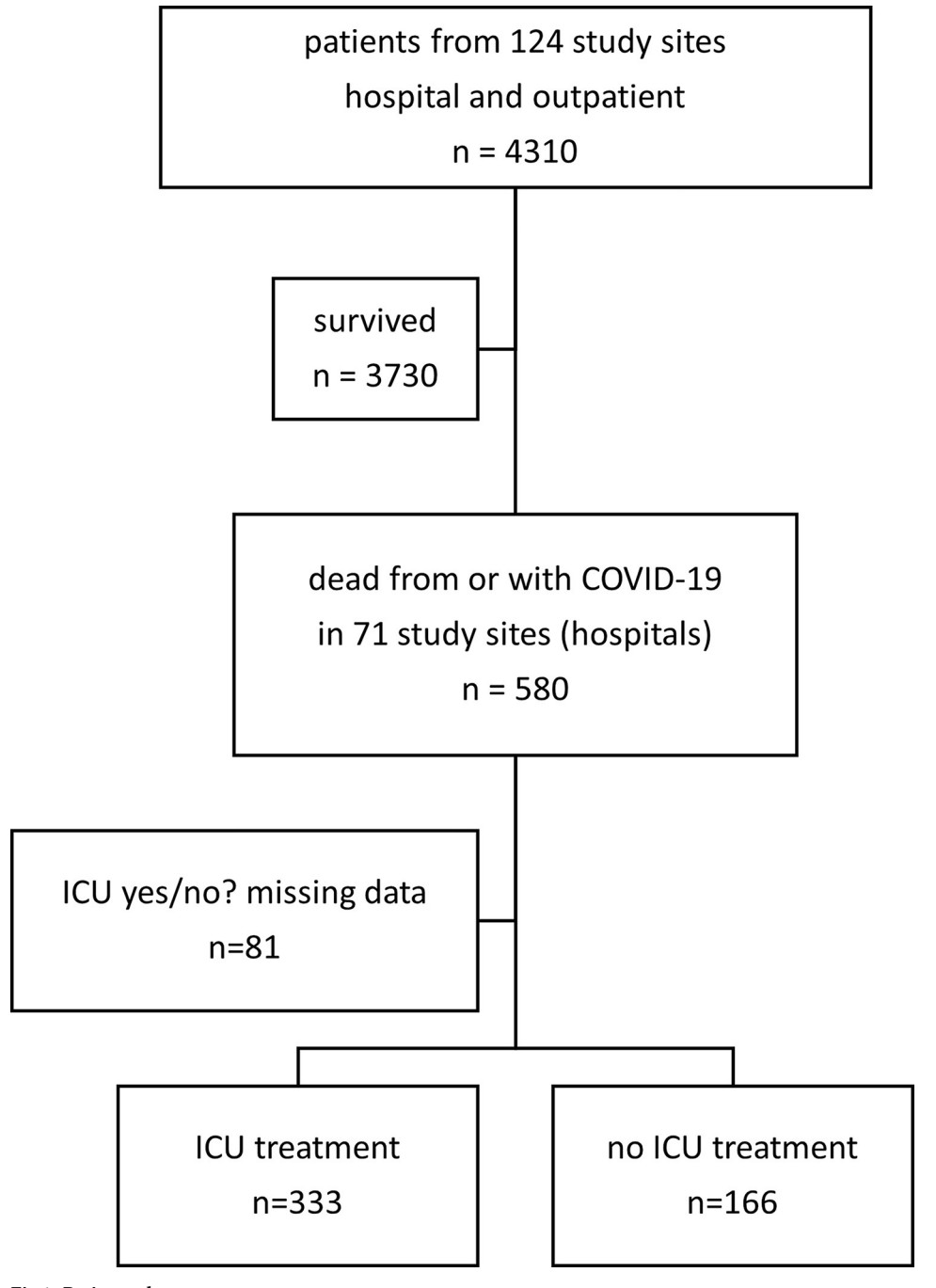

**Fig 1. Patient cohort.**

### Management of patients dying with SARS-CoV-2 infection

While about seven out of ten patients admitted to ICU came from home, patients treated on the general ward came likewise from home and nursing home (about four out of ten respectively) with about a third being on palliative care before admission compared to less than 10% in the ICU population (Table 2).

Table 2 illustrates details about the clinical management and previous care.

**Table 2. Management of patients dying with SARS-CoV-2 infection treated at ICU or general ward.**

| Characteristics | all patients | ICU | general ward | |
|---|---|---|---|---|
| | n (%) | n (%) | n (%) | p-Value |
| Patient's previous place of stay | | | | |
| at home | 181/302 (60) | 129/182 (71) | 31/70 (44) | <0.0001 |
| hospital | 43/302 (14) | 32/182 (18) | 8/70 (11) | |
| nursing home | 78/302 (26) | 21/182 (12) | 31/70 (44) | |
| On palliative care before | 65/389 (17) | 16/221 (7) | 38/129 (29) | <0.0001 |
| Involvement of palliative care team | 60/413 (15) | 26/232 (11) | 27/137 (20) | **0.031** |
| Personal contacts with family / friends before or during death phase | 221/361 (61) | 141/208 (68) | 48/103 (47) | **0.0005** |
| Change of therapeutic goals from curative to palliative | | | | |
| progress of COVID-19 only | 264/379 (70) | 164/216 (76) | 78/126 (62) | **0.031** |
| progress of COVID-19 + preexisting disease | 38/379 (10) | 18/216 (8) | 13/126 (10) | |
| progress of COVID-19 + other | 10/379 (3) | 6/216 (3) | 4/126 (3) | |
| progress of preexisting disease and/or other reasons | 67/379 (18) | 28/216 (13) | 31/126 (25) | |

Overall, involvement of a palliative care team before death of patients with SARS-CoV-2 infection was low with 15% and comparable with palliative care before admission to hospital with SARS-CoV-2 infection. Nevertheless, these groups only showed a partial overlap with about two thirds receiving palliative care before but not when hospitalized with SARS-CoV-2 infection. Personal contacts with family and/or friends before death were limited with no personal contacts documented in about 40% of patients and a lower rate for patients treated on the general ward (about half) compared to ICU (about two thirds).

## Discussion

We present a comprehensive analysis of SARS-CoV-2 infected patients having died from a European cohort focusing on clinical characteristics and in-hospital management comparing treatment on ICU versus general ward.

Our main findings were as follows: (i) About a third of patients died on a general ward and 6% of patients were referred from ICU to the general ward before death. (ii) A great majority of patients suffered from a variety of comorbidities and symptoms before death. (iii) Patients dying on the general ward were significantly older, featured more comorbidities and suffered more often from tiredness, muscle weakness and nausea/vomiting. (iv) Personal contacts were rare and even less frequently for patients treated on general ward.

Patient characteristics in our cohort were similar to findings from previous studies focusing on patients dying after infection with SARS-CoV-2. Death occurred predominantly in men and elderly patients [14, 15]. In accordance with previous studies, dying SARS-CoV-2 infected patients often suffer from comorbidities. 83% of our cohort suffered from cardiovascular diseases with hypertension as the leading disease. Cardiovascular comorbidities were followed by diabetes and kidney disease as leading comorbidities [14, 16, 17]. Our study adds important information with our detailed analysis of symptoms capturing eleven different symptoms which occurred in > 10% of the patients. Other large cohorts only list few symptoms [18]. In line with other studies, dyspnea and fever were the leading symptoms in the dying [9], but frequencies were much higher compared to other cohorts of SARS-CoV-2 infected patients (dyspnea: 77% of the dying compared to 25.5% of patients with chronic kidney diseases to 23.9% of patients with cancer; fever: 78% compared to 34% in patients with cancer) [13, 19].

Furthermore, reliable data to compare the frequency of comorbidities and symptoms of dying patients treated on ICU or general wards are missing since previous studies did not differentiate between different settings.

Our study also demonstrates that health care professionals frequently face a broad range of symptoms in SARS-CoV-2 infected patients before death irrespective of the setting. The leading symptoms are, as already mentioned, fever and dyspnoea but other symptoms such as dry cough, excessive tiredness, and delirium were also frequently observed in our cohort. The management of the wide spectrum of symptoms often demands palliative care expertise [20, 21]. Although two thirds of patients died on ICU, specialist palliative care was only involved in every tenth dying patient, 20% of dying patients on the general ward received specialist palliative care. In comparison, in a retrospective cross-sectional study of cancer patients the hospital palliative care team was involved in 30% of the patients [22].

Treatment decisions might be a particular challenge as patients often deteriorate rapidly early after admission [23]. On the day of diagnosis, 75% of our patients treated in ICU were in a complicated or critical phase of disease or died. This implies that changes in treatment goals may be necessary in many patients which could be supported by specialist palliative care. Accordingly, the need for an emergency palliative care team is postulated by Fuis-Schmidhauser et al. [24].

During the SARS-CoV-2 pandemic, isolation measures had a strong impact on patient care and restricted visits to a large extent—also for the dying [25, 26]. Data from the national Swedish registry of palliative care demonstrated the negative effect of COVID-19 on end-of-life discussions and dying without visits [27]. In our study, a relevant proportion of patients had no contacts before dying–surprisingly to a higher extent on the general ward (40%) compared to ICU (24%). Wallace et al. described how practices to restrict the spread of SARS-CoV-2 can impact on complicated grief for family members [28]. In another study, interviews with bereaved relatives indicated the distress and helplessness without the option for a personal farewell [29]. Therefore, this has been included in a national strategy developed for Germany to improve the care for severely ill and dying patients and their relatives during pandemics [30].

## Strengths

The main strength of our study is a reasonable study size with nearly 600 SARS-CoV-2 infected patients having died including a broad spectrum of comorbidities and symptoms considering the circumstances and place of death.

## Limitations

Our analysis included patients from the LEOSS cohort including European study sites. Nevertheless, 97% of our patients were from Germany, therefore results cannot easily be generalized. In this cohort, eCRFs are regularly updated and extended as knowledge related to the pandemic grows leading to a rather high missing rate for a subset of questions. Our data derive to a high extent from the beginning of the pandemic–with 88% of the patients recruited in March and April 2020 during the first pandemic wave. Whether data being obtained from patients suffering from SARS-CoV-2 infection later in the course of the pandemic differ is unknown but might be interesting to investigate. Determining the prevalence of advance directives of patients with serious illness leading to death was not possible as this item was only included later in LEOSS but will be evaluated in further studies when data are available [31].

## Conclusions

A high demand for palliative care expertise with early involvement seems to exist as dying SARS-CoV-2 infected patients often exhibit a high symptom burden and deteriorate early at the time of diagnosis with demand for ICU treatment. Moreover, palliative care specialists should be trained in care for patients on ICU as patients often die there. More effort should be made to prevent lonely dying in hospital both on ICU and on general wards to prevent distress and helplessness of patients and complicated grief of relatives.

## Acknowledgments

We express our deep gratitude to all study teams supporting the LEOSS study. The LEOSS study group contributed at least 5 per mille to the analyses of this study: Hospital Ingolstadt (Stefan Borgmann), University Hospital Freiburg (Siegbert Rieg), Hospital Ernst von Bergmann (Lukas Tometten), Technical University of Munich (Christoph Spinner), University Hospital Jena (Maria Madeleine Rüthrich), University Hospital Regensburg (Frank Hanses), University Hospital Wuerzburg (Nora Isberner), Klinikum Dortmund gGmbH (Martin Hower), Municipal Hospital Karlsruhe (Christian Degenhardt), Hospital Bremen-Center (Bernd Hertenstein), University Hospital Frankfurt (Maria Vehreschild), University Hospital Augsburg (Christoph Römmele), Elbland Hospital Riesa (Joerg Schubert), Tropenklinik Paul-Lechler-Krankenhaus Tübingen (Claudia Raichle), Hospital Maria Hilf GmbH Moenchengladbach (Juergen vom Dahl), University Hospital Erlangen (Richard Strauss), University Hospital Heidelberg (Uta Merle), Johannes Wesling Hospital Minden Ruhr University Bochum (Kai Wille), Hospital Passau (Martina Haselberger), Petrus Hospital Wuppertal (Sven Stieglitz), University Hospital Essen (Sebastian Dolff), University Hospital Ulm (Beate Gruener), Evangelisches Hospital Saarbruecken (Mark Neufang), Robert-Bosch-Hospital Stuttgart (Katja Rothfuss), Catholic Hospital Bochum (St. Josef Hospital) Ruhr University Bochum (Kerstin Hellwig), University Hospital Saarland (Robert Bals), University Hospital Cologne (Norma Jung), University Hospital Schleswig-Holstein Kiel (Anette Friedrichs), University Hospital Tuebingen (Siri Göpel), Hospital Braunschweig (Jan Kielstein), University Hospital Dresden (Katja de With), Bundeswehr Hospital Koblenz (Dominic Rauschning), Marien Hospital Herne Ruhr University Bochum (Timm Westhoff), University Hospital Munich/ LMU (Michael von Bergwelt-Baildon), Hospital Leverkusen (Lukas Eberwein), Malteser Hospital St. Franziskus Flensburg (Milena Milovanovic), University Hospital of Giessen and Marburg (Janina Trauth), University Hospital Duesseldorf (Bjoern-Erik Jensen).

The LEOSS study infrastructure group: Jörg Janne Vehreschild (Goethe University Frankfurt), Carolin E. M. Jakob (University Hospital of Cologne), Lisa Pilgram (Goethe University Frankfurt), Melanie Stecher (University Hospital of Cologne), Max Schons (University Hospital of Cologne), Susana Nunes de Miranda (University Hospital of Cologne), Clara Bruenn (University Hospital of Cologne), Nick Schulze (University Hospital of Cologne), Sandra Fuhrmann (University Hospital of Cologne), Annika Claßen (University Hospital of Cologne), Bernd Franke (University Hospital of Cologne), Fabian Praßer (Charité, Universitätsmedizin Berlin) and Martin Lablans (University Medical Center Mannheim).

## Author Contributions

**Conceptualization:** Claudia Raichle, Claudia Bausewein, Steffen T. Simon, Charlotte Leisse, Berenike Pauli, Norma Jung.

**Data curation:** Carolin E. M. Jakob, Jörg Janne Vehreschild, Melanie Stecher, Lisa Pilgram.

**Formal analysis:** Claudia Raichle, Norma Jung.

**Funding acquisition:** Jörg Janne Vehreschild.

**Investigation:** Claudia Raichle, Claudia Bausewein, Siegbert Rieg, Carolin E. M. Jakob, Steffen T. Simon, Lukas Tometten, Jörg Janne Vehreschild, Charlotte Leisse, Johanna Erber, Melanie Stecher, Berenike Pauli, Maria Madeleine Rüthrich, Lisa Pilgram, Frank Hanses, Nora Isberner, Martin Hower, Christian Degenhardt, Bernd Hertenstein, Maria J. G. T. Vehreschild, Christoph Römmele, Norma Jung.

**Project administration:** Carolin E. M. Jakob, Jörg Janne Vehreschild, Melanie Stecher, Lisa Pilgram.

**Resources:** Claudia Raichle, Stefan Borgmann, Claudia Bausewein, Siegbert Rieg, Carolin E. M. Jakob, Steffen T. Simon, Lukas Tometten, Jörg Janne Vehreschild, Charlotte Leisse, Johanna Erber, Melanie Stecher, Berenike Pauli, Maria Madeleine Rüthrich, Lisa Pilgram, Frank Hanses, Nora Isberner, Martin Hower, Christian Degenhardt, Bernd Hertenstein, Maria J. G. T. Vehreschild, Christoph Römmele, Norma Jung.

**Writing – original draft:** Claudia Raichle, Norma Jung.

**Writing – review & editing:** Claudia Raichle, Stefan Borgmann, Claudia Bausewein, Siegbert Rieg, Carolin E. M. Jakob, Steffen T. Simon, Lukas Tometten, Jörg Janne Vehreschild, Charlotte Leisse, Johanna Erber, Melanie Stecher, Berenike Pauli, Maria Madeleine Rüthrich, Lisa Pilgram, Frank Hanses, Nora Isberner, Martin Hower, Christian Degenhardt, Bernd Hertenstein, Maria J. G. T. Vehreschild, Christoph Römmele, Norma Jung.

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
