## [Decision Letter · Decision Letter 0]

31 Jan 2022

PONE-D-21-32351Hospitalized patients dying with SARS-CoV-2 infection – an analysis of patient characteristics and management in ICU and general ward of the LEOSS registryPLOS ONE

Dear Dr. Norma Jung,

Thank you for submitting your manuscript to PLOS ONE. After careful consideration, we feel that it has merit but does not fully meet PLOS ONE’s publication criteria as it currently stands. Therefore, we invite you to submit a revised version of the manuscript that addresses the points raised during the review process.

We look forward to receiving your revised manuscript.

Kind regards,

Paavani Atluri

Academic Editor

PLOS ONE

Journal Requirements:

2. Please include your full ethics statement in the ‘Methods’ section of your manuscript file. In your statement, please include the full name of the IRB or ethics committee who approved or waived your study, as well as whether or not you obtained informed written or verbal consent. If consent was waived for your study, please include this information in your statement as well

Reviewers' comments:

Reviewer's Responses to Questions

**Comments to the Author**

1. Is the manuscript technically sound, and do the data support the conclusions?

Reviewer #1: Yes

Reviewer #2: Yes

2. Has the statistical analysis been performed appropriately and rigorously? 

Reviewer #1: Yes

Reviewer #2: No

3. Have the authors made all data underlying the findings in their manuscript fully available?

Reviewer #1: Yes

Reviewer #2: No

4. Is the manuscript presented in an intelligible fashion and written in standard English?

Reviewer #1: Yes

Reviewer #2: Yes

5. Review Comments to the Author

Reviewer #1: Interesting study with really good number of patients. It would be interesting to find out the data in the current vaccination era. Palliative consult is a must in all ICU patients, involving them early is very crucial.

My revisions include:

In the study design - Quotations before "Lean Europe survey (line 96)

Table 1: in the BMI group, please replace decimal points instead of comma

Reviewer #2: During this COVID-19 pandemic, In-hospital mortality among patients with COVID-19 was reportedly high. It is imperative to study the characteristics of Hospitalized patients dying with SARS-CoV-2 infection. This multicentre analysis has compared and highlighted some of these factors in ICU and general ward from the LEOSS registry. This retrospective data analysis has presented some important findings. Moreover, some of the queries related to the manuscript has been listed below:

1. In the Method section, the authors have mentioned it was a ‘Prospective’ study. However, it looks like a ‘Retrospective’ study. There is a discrepancy on page 5, line no 97 and line no. 107. The author should explain and correct it.

2. Why authors have selected specific dates for the analysis, i.e. March 18 and November 18, 2020. Also, the authors have stated most of the patients (~80%) were in March and April 2020. Is it in line with the first wave in the country? The authors may explain.

3. Page 5, line no 116 was unclear if ‘Table 1 referred to Rüthrich et al. or present study. Rewrite the sentence.

4. Page 7, line no 147, was about Bonferroni correction. The author should calculate and report the correct significant level as the number of tests ~28, <0.01 was inadequate.

5. Lastly, post hoc analysis would also be useful to report, if possible.

6. PLOS authors have the option to publish the peer review history of their article (what does this mean?). If published, this will include your full peer review and any attached files.

Reviewer #1: No

Reviewer #2: **Yes: **Prabal Kumar Chourasia

---

## [Author Response · Author response to Decision Letter 0]

18 Feb 2022

Dear Dr. Atluri,

on behalf of all authors, I would like to thank the Editors and the Reviewers for the critical remarks on our manuscript (PONE-D-21-32351: Hospitalized patients dying with SARS-CoV-2 infection – an analysis of patient characteristics and management in ICU and general ward of the LEOSS registry) we received on the 1st February. We are grateful to get the opportunity to resubmit our revised manuscript. We very much appreciate the comments which we considered in the revised version and which we found helpful to improve the clarity of our manuscript.

All answers to the comments are highlighted in yellow. 

We hope that the paper is now of sufficient quality to be published in Plos One.

Yours sincerely,

Norma Jung, MD

Reviewers' comments:

Reviewer #1: Interesting study with really good number of patients. It would be interesting to find out the data in the current vaccination era. Palliative consult is a must in all ICU patients, involving them early is very crucial.

My revisions include:

In the study design - Quotations before "Lean Europe survey (line 96)

Thanks – we skipped the Quotations

Table 1: in the BMI group, please replace decimal points instead of comma

Thanks – we changed to points. 

Reviewer #2: During this COVID-19 pandemic, In-hospital mortality among patients with COVID-19 was reportedly high. It is imperative to study the characteristics of Hospitalized patients dying with SARS-CoV-2 infection. This multicentre analysis has compared and highlighted some of these factors in ICU and general ward from the LEOSS registry. This retrospective data analysis has presented some important findings. Moreover, some of the queries related to the manuscript has been listed below:

1. In the Method section, the authors have mentioned it was a ‘Prospective’ study. However, it looks like a ‘Retrospective’ study. There is a discrepancy on page 5, line no 97 and line no. 107. The author should explain and correct it.

Thanks for this helpful remark – patients were recruited prospectively and data documented retrospectively. For more clarity, we scipped the word prospective in the method section. 

Line 97: “Data were retrieved from the Lean European Open Survey on SARS-CoV-2 infected patients (LEOSS) registry, a multi-center non-interventional cohort study.” 

2. Why authors have selected specific dates for the analysis, i.e. March 18 and November 18, 2020. Also, the authors have stated most of the patients (~80%) were in March and April 2020. Is it in line with the first wave in the country? The authors may explain.

Thanks for your question. Start day of our study (March 18) is the start day of the LEOSS-cohort, end day of our study (November 18) was set to the date of data retrieval from the ongoing cohort study, following this, we started with our analysis. 

Yes -March and April 2020 were the time of the first wave in the country- we included this information in the discussion section-chapter discussion – limitations

Line 266: “Our data derive to a high extent from the beginning of the pandemic – with 88% of the patients recruited in March and April 2020, during the first pandemic wave.”

3. Page 5, line no 116 was unclear if ‘Table 1 referred to Rüthrich et al. or present study. Rewrite the sentence.

Thanks for this remark. For more clarity we rewrote the sentence from: “For more details see Rüthrich et al.(1) , Table 1.” into “For more details see Table 1 in the study of Rüthrich et al. (1). (Line 116)

4. Page 7, line no 147, was about Bonferroni correction. The author should calculate and report the correct significant level as the number of tests ~28, <0.01 was inadequate.

Thanks for your accurate review. Please excuse the inattentiveness, this sentence suggesting multiple testing was supposed to be deleted in the current version, as the statistical methods used do not require a Bonferroni adjustment. We performed univariate analyses with the different parameters in one dataset(same patient groups, each parameter is tested once, no multiple testing, no post-hoc analysis).

We skipped the sentence and inserted instead: “Reported p values are 2-sided and p<0.05 was considered statistically significant.” . We adjusted the highlighting of the statistically significant parameters in the tables.

5. Lastly, post hoc analysis would also be useful to report, if possible.

Thanks for your advice. We already answered your question in the previous section (4).

---

## [Decision Letter · Decision Letter 1]

15 Jun 2022

PONE-D-21-32351R1Hospitalized patients dying with SARS-CoV-2 infection – an analysis of patient characteristics and management in ICU and general ward of the LEOSS registryPLOS ONE

Dear Dr. Jung,

Thank you for submitting your manuscript to PLOS ONE. After careful consideration, we feel that it has merit but does not fully meet PLOS ONE’s publication criteria as it currently stands. Therefore, we invite you to submit a revised version of the manuscript that addresses the points raised during the review process.

We look forward to receiving your revised manuscript.

Kind regards,

Paavani Atluri

Academic Editor

PLOS ONE

Journal Requirements:

Reviewers' comments:

Reviewer's Responses to Questions

**Comments to the Author**

1. If the authors have adequately addressed your comments raised in a previous round of review and you feel that this manuscript is now acceptable for publication, you may indicate that here to bypass the “Comments to the Author” section, enter your conflict of interest statement in the “Confidential to Editor” section, and submit your "Accept" recommendation.

Reviewer #2: All comments have been addressed

2. Is the manuscript technically sound, and do the data support the conclusions?

Reviewer #2: Yes

3. Has the statistical analysis been performed appropriately and rigorously? 

Reviewer #2: Yes

4. Have the authors made all data underlying the findings in their manuscript fully available?

Reviewer #2: Yes

5. Is the manuscript presented in an intelligible fashion and written in standard English?

Reviewer #2: Yes

6. Review Comments to the Author

Reviewer #2: Authors have addressed all the comments in the main text. A couple of minor edits are suggested in the abstract to align with the main text listed below:

1. Abstract Page 4 line 60 Remove ‘prospective’ term

2. Abstract Page 4 line 65 second sentence is unclear if 67% of 4310 or 67% of 580. Suggest to add ‘Among 580 patients,’ before the text.

I have no further comments in the MS.

7. PLOS authors have the option to publish the peer review history of their article (what does this mean?). If published, this will include your full peer review and any attached files.

Reviewer #2: **Yes: **Prabal Chourasia

---

## [Author Response · Author response to Decision Letter 1]

22 Jun 2022

Reviewers' comments:

Reviewer #2: Authors have addressed all the comments in the main text. A couple of minor edits are suggested in the abstract to align with the main text listed below:

1. Abstract Page 4 line 60 Remove ‘prospective’ term

Thanks – we removed “prospective”.

2. Abstract Page 4 line 65 second sentence is unclear if 67% of 4310 or 67% of 580. Suggest to add ‘Among 580 patients,’ before the text.

Thanks for your helpful remark – we added “among 580 patients”.

I have no further comments in the MS.

---

## [Editor Report · Decision Letter 2]

8 Jul 2022

Hospitalized patients dying with SARS-CoV-2 infection – an analysis of patient characteristics and management in ICU and general ward of the LEOSS registry

PONE-D-21-32351R2

Dear Dr. Jung,

We’re pleased to inform you that your manuscript has been judged scientifically suitable for publication and will be formally accepted for publication once it meets all outstanding technical requirements.

Kind regards,

Paavani Atluri

Academic Editor

PLOS ONE
---

## [Editor Report · Acceptance letter]

21 Jul 2022

PONE-D-21-32351R2 

Hospitalized patients dying with SARS-CoV-2 infection – an analysis of patient characteristics and management in ICU and general ward of the LEOSS registry 

Dear Dr. Jung:

I'm pleased to inform you that your manuscript has been deemed suitable for publication in PLOS ONE. Congratulations! Your manuscript is now with our production department. 

Kind regards, 

on behalf of

Dr. Paavani Atluri 

Academic Editor

PLOS ONE